# Environmental justice and power plant emissions in the Regional Greenhouse Gas Initiative states

**Juan Declet-Barreto** [1]* , **Andrew A. Rosenberg** [2]

**1** Climate & Energy Program, Union of Concerned Scientists, Washington, DC, United States of America,
**2** Center for Science and Democracy, Union of Concerned Scientists, Cambridge, MA, United States of America

☉ These authors contributed equally to this work.

* jdeclet-barreto@ucsusa.org

**Data Availability Statement:** https://osf.io/tu2hj/.

**Funding:** Juan Declet-Barreto's contribution came from work completed while a Kendall Science Fellow at the Union of Concerned Scientists (UCS),

## Abstract

Policies to reduce greenhouse gases associated with electricity generation have been a major focus of public policy in the United States, but their implications for achieving environmental justice among historically overburdened communities inappropriately remains a marginal issue. In this study we address research gaps in historical and current ambient air emissions burdens in environmental justice communities from power plants participating in the Regional Greenhouse Gases Initiative (RGGI), the country's first market-based power sector emissions reduction program. We find that in RGGI states the percentage of people of color that live within 0–6.2 miles from power plants is up to 23.5 percent higher than the percent of the white population that lives within those same distance bands, and the percentage of people living in poverty that live within 0–5 miles from power plants is up to 15.3 percent higher than the percent of the population not living in poverty within those same distance bands. More importantly, the transition from coal to natural gas underway before RGGI formally started resulted in large increases in both the number of electric-generating units burning natural gas and total net generation from natural gas in environmental justice communities hosting electric-generating units, compared to other communities. Our findings indicate that power sector carbon mitigation policies' focusing on aggregate emissions reductions have largely benefitted non-environmental justice communities and have not redressed the fundamental problem of disparities in pollutant burdens between EJ and non-EJ communities. These must be directly addressed in climate change and carbon emissions mitigation policy.

## Introduction

The electricity sector accounts for 29 percent of total U.S. greenhouse gas (GHG) emissions and about 32% of total U.S. energy-related carbon dioxide ($CO_2$) emissions [1, 2]. Almost all of the U.S. electricity sector's $CO_2$ emissions (98%) come from the burning of coal and natural

which is funded by unrestricted dollars allocated by the UCS Board. The funders had no role in study design, data collection and analysis, decision to publish, or preparation of the manuscript.

**Competing interests:** The authors have declared that no competing interests exist.

gas. Thus, reducing GHGs associated with electricity generation has been a major focus of public policy to address climate change at federal, regional and state levels across the US.

The fossil-fuel based energy sector remains a priority for climate change mitigation efforts. Historically, concern about the intersection of climate change mitigation and social equity has been a marginal issue in the domestic U.S. policy agenda [3]. The issue of equity has often focused on the question of potential unequal rate-payer cost burdens of various regulatory schemes [e.g., 4]. Nevertheless, there are other equity issues that concern environmental justice scholars and advocates. Prime among these is the problem of co-pollutant emissions [5].

In addition to emitting carbon dioxide—a globally-dispersed pollutant—the power sector is responsible for emissions of other locally or regionally-dispersed pollutants. Electricity generation constitutes 64 percent of $SO_2$ emissions, 14 percent of NOx emissions, 3.4 percent of $PM_{2.5}$ emissions, and 1.4 percent of $PM_{10}$ emissions in the U.S. [6]. Co-pollutants from electricity generation are responsible for significant health impacts on local communities, and contribute to the disproportionate health-impairing pollution burden that exists in many environmental justice communities [7–9].

This paper reports the results of an empirical investigation of $CO_2$ and two co-pollutant (NOx, $SO_2$) emissions from electricity generation in states participating in the Regional Greenhouse Gas Initiative (RGGI). RGGI is the United States' first market-based emissions reduction program, established in 2009 with the intent of reducing carbon emissions from electric generation in those states [10, 11].

The purpose of this study is to address the research gaps in historical and current ambient air emissions burdens from power plants in environmental justice communities. We identified large differences in siting and operation of power plants between communities of color and low-income communities, versus other communities. As policies to mitigate climate change and pollution move forward they must directly address the heterogeneity of conditions in communities of color and low income versus other communities in order to be both just and effective.

## Electricity sector emissions and environmental justice

The burden of mortality and morbidity due to exposure to co-pollutants associated with electricity generation (e.g., NOx, $SO_2$, $PM_{10}$ and $PM_{2.5}$) is well established [12–16]. Reducing emissions of power plant co-pollutants can have many human health benefits, including decreases in all-cause [17] and $PM_{2.5}$-attributable mortality and morbidity; decreased hospitalizations due to myocardial infarctions and respiratory and cardiovascular disease [18–20], significant economic savings due to lost work days averted; and billions of dollars in health savings and other health benefits [21]. In the United States, a large percentage of both low-income populations and people of color live close to power plants [6, 22, 23], making equitable reductions in power sector emissions a critical public health and environmental justice issue.

Addressing the disproportionate burdens from ambient air emissions and other environmental risks in Black, Indigenous, communities of color and low-income communities is the focus of environmental justice scholars and advocates [24–26]. Achieving environmental justice is the goal of fairly and equitably addressing the fact that some communities, which are typically low-income communities of color, are overburdened with environmental and economic burdens [27].

While there is a substantial body of research linking air quality and health, studies examining the *distribution* of such benefits across communities are much less abundant. Are the benefits from GHG and co-pollutant reductions distributed equitably across communities? Or do some communities benefit more than others? The distribution question is at the center of

environmental justice and climate mitigation research. Environmental justice scholars and advocates contend that mitigation policies should address the attendant co-pollutant emissions associated with fossil fuel combustion in environmental justice communities, thereby also reducing disparities in existing environmental hazard burdens [e.g., 5]. Environmental justice advocates have largely not supported market-based solutions for emissions reductions because there is no explicit recognition nor requirement to address equity, arguing that climate change mitigation policy provides an opportunity to address historical and current pollution burdens—an opportunity that should not be forgone [5]. Environmental justice advocates maintain that in excluding race- and income-based equity considerations–the fundamental historical problem in environmental justice—efforts to reduce GHGs could fail to reduce emissions in environmental justice communities that already face substantial cumulative environmental hazards burdens [28, 29]. Others have found that energy sector decarbonization policies that explicitly incorporate both the realization of environmental justice and improved air quality result in substantial reductions in damages from co-pollutants [30]. Another concern is the possibility that emissions "hotspots" may emerge from local, large point sources like power plants, increasing co-pollutant exposures in overburdened communities. If, for example, the costs of emissions reductions in plants located in environmental justice communities are higher than in plants located elsewhere, market logic will lead to increases in carbon and co-pollutant emissions in those communities as utilities pursue lower-cost reductions elsewhere [3]. Carbon-trading programs, environmental justice scholars contend, are ill-suited to pre-vent local (i.e., at a specific particular power plant) emissions increases because these programs establish overall emissions reductions goals, and allow regulated entities to trade allowances with each other as a primary compliance mechanism [31]. Environmental justice advocates also reject that climate and air pollution should be handled separately, and warn against miss-ing the opportunity to use climate change mitigation policy to achieve reductions in power plant co-pollutants that have not been realized under the Clean Air Act [5]. In addition, others have voiced more fundamental problems with carbon trading that do not center on emissions reductions, namely that carbon markets undermine efforts to decarbonize the global economy [32].

The analysis presented here investigates empirically carbon and co-pollutant emissions associated with electricity generating units (EGUs) in the ten states participating in RGGI: Connecticut, Delaware, Maine, Massachusetts, Maryland, New Hampshire, New Jersey, New York, Rhode Island, and Vermont. New Jersey was an initial participant, withdrew in 2012, and in 2018 its newly-elected governor issued an executive order to rejoin. We selected these states because as subjects of the first regional climate mitigation initiative in the electricity sec-tor in the United States, they offer an interesting starting point for assessing distributive out-comes of carbon and co-pollutant reductions between EJ and non-EJ communities.

This study is a longitudinal exploration of ambient air emissions originating from power plants located in environmental justice communities and communities not considered envi-ronmental justice communities. We developed the following guiding questions, in consulta-tion with some leaders in the environmental justice movement, as an exploratory first step toward understanding electricity sector emissions in environmental justice communities:

1. To what extent are communities of color and low-income communities overrepresented in residential proximity to power plants in the RGGI states?

2. Are income and race an appropriate basis for a valid and reliable quantitative delineation of environmental justice (EJ) communities versus non-EJ communities, suitable for longitudi-nal analyses of power plant emissions?

3. What is the distribution of the polluting potential as measured by fuel type, total net generation, capacity factor, and of actual pollution from emissions of $CO_2$, $SO_2$, and $NO_x$ from electric-generating units (EGUs) sited in environmental justice communities compared to EGUs sited in non-environmental justice communities?

4. What changes in the relationships between power plant characteristics and community types (EJ versus non-EJ) have occurred since 2009?

To answer these questions, we first develop empirical cumulative distribution functions (CDFs) of population groups indicative of environmental justice community status versus distance to power plants. We then group each census tract in our study area into one of two clusters of high vs low fractions of people of color and people living in poverty, which we label environmental justice communities (EJC) or non-environmental justice communities (non-EJCs). We then estimate six indicators of power plants' potential to pollute (siting frequency, electricity generation capacity, capacity factor, and emissions of CO2, SO2, and NOx), stratifying all indicators by reported fuel type (oil, coal, biomass, or natural gas) separately for power plants sited in EJC vs. non-EJCs. We conclude with a discussion of the implications of our findings for the incorporation of environmental justice goals for overburdened communities in power sector carbon emissions reduction policy.

## Data

### Database of historical emissions from EGUs

An EGU is a fossil fuel-fired combustion unit that serves a generator that produces electricity for sale [33]. We selected EGUs as the unit of analysis because fuel type data are aggregated at the EGU level, a key component of our analysis. The U.S. Energy Information Administration (EIA) administers yearly surveys to electrical power producers. Survey form 860 provides information on all EGUs that are currently in operation or have retired since 2001, including data on the generator's latitude and longitude, first and last years of operation, and information on installed environmental controls. Form 923 provides data on electricity production and fuel consumption. Emissions data for $CO_2$, NOx, and $SO_2$, generation, fuel inputs, and environmental controls were provided by the EPA's Air Markets Program [34]. Based on these publicly available data, we obtained a combined database of all EGUs in the United States, along with data describing historical electricity generation, capacity, emissions, and other attributes for the period 1995–2015 [34–37]. Due to gaps in EGU data reporting for the years 2001 and 2002, we did not use data from those years in our analyses. We focused on EGUs with rated (or nameplate) capacity of 25 Megawatts (MW) or higher located in RGGI states.

There were 333 EGUs with a rated capacity of 25 MW or higher in RGGI states. There were 261 individual facilities ("fencelines"), and some facilities host more than one EGU; 41 of these facilities co-hosted between two and three EGUs. We eliminated 23 EGUs without valid latitude/longitude coordinates. In our final selection we included 310 EGUs, or 93 percent of all eligible EGUs (Fig 1).

### Race/ethnicity and poverty in classifying environmental justice communities

Socio-demographic data on poverty status and race/ethnicity by census tract was obtained for 2010 from the U.S. Census Bureau [38, 39]. Two indicators commonly used to identify EJCs were calculated: First, "Percent people of color" was defined as the percentage of the total population in a census tract that identifies as non-Hispanic African American, American Indian, Alaska Native, Asian, Pacific Islander, other non-White race, or of Hispanic/Latino/Latina

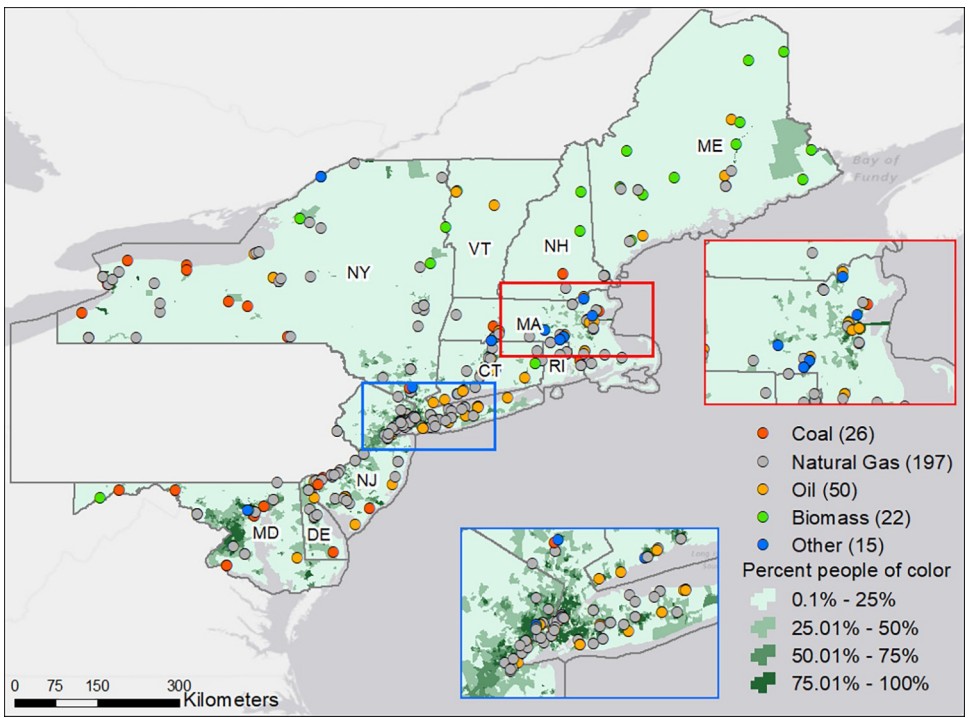

**Fig 1. Electric Generating Units (EGUs) and people of color in RGGI states.** Each EGU is symbolized according to its most-commonly used fuel in 2015.

origin of any race or races. "Percent living in poverty" was defined as the percent of individuals with a ratio of income to poverty level below one, that is, individuals with incomes below the federal poverty threshold.

## Analysis

### Population proximity to power plants

We estimated disparities in the proximity of populations to EGUs by developing empirical cumulative distribution functions (CDFs) of population versus distance to EGUs. CDFs of population proximity to industrial point sources have been proposed as providing more meaningful representation of distance-based exposure than fixed-distance bands [40]. We first calculated the straight-line distance (in miles) of the nearest power plant to the geographical center of each census tract. Tracts were sorted in increasing order of distance, and the cumulative population at distance *j* was defined as the sum of the population in all census tracts at a distance less than or equal to distance *j* from power plants. Distance was then plotted, separately for each of four population subgroups against the percentage of each subpopulation out of the total population at that distance. The four population subgroups included were whites, people of color, the population below the poverty threshold, and the population at or above the poverty threshold. For the purposes of assessing population proximity to power plants, we deemed people of color and the population below the poverty threshold as environmental justice populations, and whites and the population at or above the poverty threshold as reference, non-environmental justice populations. We did not consider the intersection of low-income and white populations as environmental justice communities, instead focusing on mutually exclusive population subgroup pairs of whites/people of color; and persons living below, or at/above the poverty threshold.

## Identification of environmental justice communities

While defining environmental justice communities has been a subject of research and documentation, there is no generally accepted standard research definition of an environmental justice community. Some studies utilize fine-scale census data on the racial/ethnic composition and poverty levels of populations to assess the environmental justice status of communities [26, 41–43]. Others incorporate social, economic, and environmental indicators such as economic inequality [44, 45]; vulnerability based on age [46], political enfranchisement and educational achievement [47, 48], level of civic engagement [26], linguistic isolation [49], sexual orientation and gender identity [50], and criteria air pollution [51].

There is some regulatory guidance on how to identify environmental justice communities [22, 52, 53]; however, the application of such guidance to systematically and quantitatively identify environmental justice communities does not exist. In our research, we address this gap by developing and applying a cluster analysis to identify Census tracts as EJCs or non-EJCs. In our research, EJCs are defined solely by race, ethnicity, and poverty indicators. In our study, we deem race, ethnicity, and poverty indicators to be the best publicly and systematically-available proxies for identifying EJCs across a large study area. Our quantitative definition of EJCs may differ from other approaches to identifying specific communities as EJCs or non-EJCs. We adhered to EPA's guidance on identification of EJ communities using percent in poverty and percent people of color indicators (e.g., 22, 49, 50), and we did not explicitly separate low-income white communities from other low-communities in our analysis.

Following Collins et al. [50] we conducted a two-cluster *k*-means analysis for the 11,813 Census Tracts in RGGI states. The *k*-means technique is an algorithm used to classify observations in a dataset into *k*-number of clusters or categories, where each cluster has a mean that produces the smallest within-cluster sum of squares. Rather than inferring poverty or race/ethnicity composition thresholds, we used cluster analysis as a statistical method for grouping each census tract in our study area into one of two clusters of high vs low fractions of people of color and people living in poverty. Based on the *k*-means cluster analysis, we assigned census tracts with higher fractions of people of color and people living in poverty to the EJCs category, and census tracts with lower fractions of the same variables to the non-EJCs category. *k*-means algorithms are based on averages; as such, our method may not accurately group outliers of extreme values in race/ethnicity and poverty variables. Although our analysis of EGUs is focused only on Census Tracts hosting EGUs (and not all 11,813 Census Tracts in RGGI states), we conducted the cluster analysis on all Census Tracts in RGGI in order to obtain a classification of Census Tracts that is representative of the distribution of poverty and race/ethnicity across RGGI states. Thus, we selected the 219 Census Tracts with at least one EGU located within each tract in our study area, eliminating 8 that had incomplete Census data on race/ethnicity and poverty variables. For each tract of the selected 211 tracts, we calculated two indicators of EJ community status—percent people of color and percent living at or below the poverty threshold—and assigned one of two categories (EJC, non-EJC) to each tract based on the *k*-means cluster analysis.

## Polluting potential of EGUs

We estimated six indicators of an EGU's potential to pollute, stratifying all indicators by reported fuel type (oil, coal, biomass, or natural gas) separately for EGUs sited in EJ vs. non-EJ communities. First, we calculated the total number of EGUs sited within tracts designated as EJCs and tracts designated as non-EJCs as an initial estimation of polluting burden. Second, we used data on annual electricity generation by fuel type in the EGU database to assign a majority fuel type to EGUs based on the largest fraction of generation in each reporting year

and calculate total net generation in both EJCs and non-EJCs. Third, we combined EGU-level nameplate capacity information (in Megawatts) with generation data to calculate each EGU's capacity factor, a number that represents the ratio of actual electricity produced to the maximum possible generation of the unit. For each year and fuel type, we calculated the mean capacity factor of EGUs that reported capacity and generation data. The final three indicators of RGGI EGUs' polluting potential are the average annual $CO_2$, $SO_2$, and $NO_x$ stack emissions. We calculated all indicators for the years 1995–2015, with the exception of 2001 and 2002 as noted above.

## Results

### Population proximity to power plants

For short distances of residential proximity to power plants in RGGI, the cumulative percentage of EJ populations is markedly higher than the cumulative percentage of non-EJ populations (Fig 2). Between 0 and 6.2 miles, the percent difference between people of color and Whites increases to a maximum of 23.5% (Fig 2A). For distances > = 6.2 miles, the difference decreases gradually, amounting to <1.0% for distances > = 39.4 miles. The results for poverty are similar: between 0 and 4.9 miles, the cumulative percentage of people living in poverty is up to 15.3% higher than the cumulative percentage of people not living in poverty (Fig 2B). For distances > = 4.9 miles, the percent difference decreases gradually to <1.0% at a distance of 20.5 miles or more.

### Identification of environmental justice communities

The cluster analysis cleanly split Census Tracts into two groups; 59 census tracts were classified as EJCs and 152 tracts were classified as non-EJCs. The EJCs had, on average, 65.8 percent people of color and 19.0 percent living with low incomes. The 152 tracts classified as non-EJCs averaged 15.9 percent people of color and 10.6 percent living with low incomes (Table 1). Of the 152 tracts classified as non-EJCs, five tracts with percent people of color one standard deviation above the mean fell in the non-EJC category and are labeled as non-EJC outliers (Fig 3).

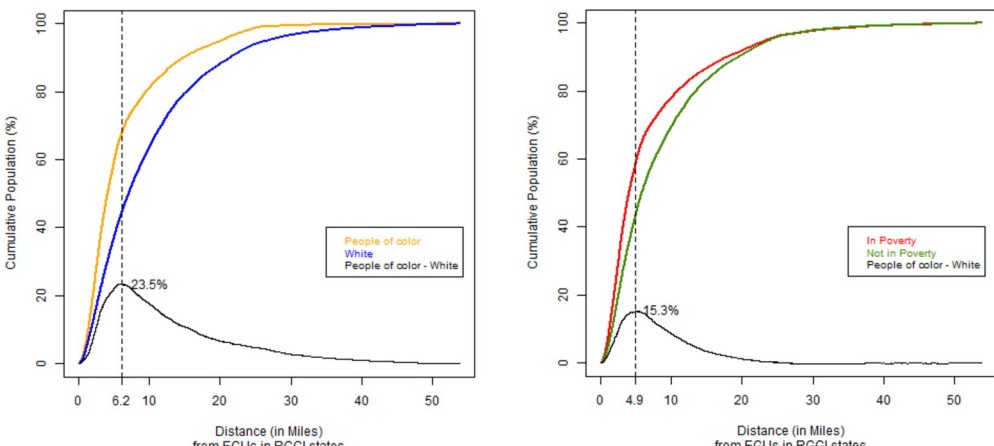

**Fig 2.** a. Cumulative distribution of people of color and White populations by distance to EGUs in RGGI. Black line indicates between the cumulative population curves. The maximum percent difference is labeled on the figure. Dotted vertical line indicates distance of maximum difference value. b. Cumulative distribution of people living in poverty and people not living in poverty by distance to EGUs in RGGI. Black line indicates difference between the cumulative population curves. The maximum percent difference is labeled on the figure. Dotted vertical line indicates distance of maximum difference value.

**Table 1. Basic statistics of k-means cluster analysis of census tracts hosting EGUs in RGGI.**

| | Percent people of color | | Percent in poverty | |
|---|---|---|---|---|
| | mean | Sd | mean | sd |
| EJCs (n = 59) | 65.8 | 16.4 | 19.0 | 14.0 |
| non-EJCs (n = 152) | 15.9 | 11.2 | 10.6 | 7.9 |

In Fig 3 we have shown only the Census Tracts hosting EGUs; see S1 Fig for the EJC/non-EJC classification of all Census Tracts in RRGI.

Of the 310 EGUs in our analysis, 73 (23.5%) were sited in Environmental Justice communities, and the remaining 237 (76.5%) were sited in non-Environmental Justice communities. When looking at the distribution of EGUs within communities, we find that EJ communities have a higher relative frequency of multiple EGUs sites when compared to non-EJ communities: among all EJ communities in our study area, 57.4% have exactly one EGU, versus the large majority of non-EJ communities, where 71.3% of all communities host exactly one EGU (Table 2). Consequently, 42.6% of EJ communities host between 2 and 5 EGUs, compared to only 28.7% in non-EJ communities.

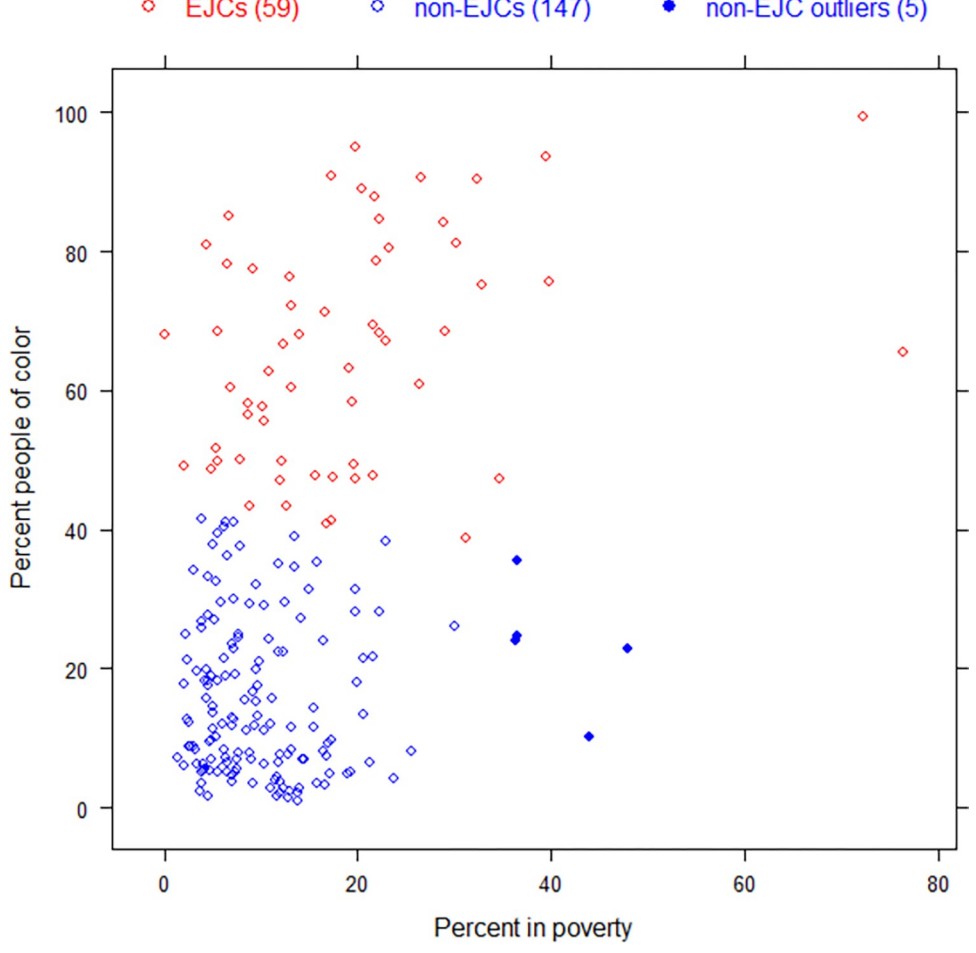

**Fig 3. Percent people of color and percent in poverty statistics in k-means cluster analysis of census tracts hosting EGUs in RGGI.**

**Table 2. Number of census tracts with one and between 2–5 EGUs sited within EJCs and non-EJCs communities in RGGI.**

| | Number of Census Tracts | |
|---|---|---|
| Number of EGUs sited within | in EJ communities | in non-EJ communities |
| 1 | 27 (57.4%) | 122 (71.3%) |
| Between 2 and 5 | 20 (42.6%) | 49 (28.7%) |

## Polluting potential of EGUs

**Number of EGUs.** The longitudinal analysis of EGUs by fuel type shows that since 1995, natural gas increasingly became a larger percentage of EGUs in both EJCs and non-EJCs (Fig 4). However, since 2003, the number of natural gas EGUs as a percent of all EGUs was higher in EJ communities than in non-EJ communities. For every year in 1995–2015, the percent of coal-fired EGUs in EJ communities was consistently smaller than that of coal-fired EGUs in non-EJ communities. The number of EGUs in each type of community also grew during the time period: from 50 to 70 (a 40.0% increase) in EJ communities and from 193 to 225 (a 16.5% increase) in non-EJ communities, and both communities saw the sharpest increases in after 2003. Biomass-burning EGUs were located only in non-EJ communities. To assess the statistical significance of changes in the number of power plants, we conducted one-way ANOVA of the frequency of oil-, natural gas-, and coal-fired power plants. We excluded biomass-fired plants because no biomass plants regulated by RGGI were sited in EJ communities in our analysis; we also excluded plants that did not report fuel type. The p-values indicate that the number of power plants of each fuel type are different in statistically-significant ways in EJCs compared to non-EJCs (S1 Table).

**Electricity generation.** Total net electricity generation in EGUs sited in both EJ and non-EJ communities has been dominated by coal and natural gas-fired units (Fig 5), with a decrease in the share of coal generation in tandem with an increase in natural gas since 1995. In EGUs sited in EJ communities, since 1995 natural gas became a larger share of total generation than in non-EJ communities. For each year in the study period, the fraction of coal-based generation has been larger in non-EJ communities than in EJ communities. In units sited in non-EJ communities, total net generation increased from about 104,358 GWh in 1995—peaking in 2005 at 180,316 GWh and then decreasing to 132,988 GWh in 2015. In EJ communities, generation increased consistently from 25,091 Gwh in 1995 to 52,529 Gwh in 2015.

**Capacity factor.** During the study period, mean capacity factors of oil and natural gas EGUs were similar in both EJCs and non-EJCS (Fig 6). Coal-fired EGUs in EGUs sited in non-EJCs have higher capacity factors than in those sited in EJCs. As in Figs 4 and 5, biomass-fired EGUs have been present only in non-EJCs.

**Emissions of $CO_2$, $SO_2$, and NOx.** The longitudinal trajectory of average annual $CO_2$ emissions was generally similar for EGUs located in EJ communities and in non-EJ communities (Fig 7), but there are some important differences. First, between 1995–2005, $CO_2$ emissions in EJCs were slightly higher. While EGUs in both types of communities saw sustained reductions in $CO_2$ emissions after 2005, those reductions were larger in EJCs after 2009 (with the exception of an increase in 2010). After 2003, $CO_2$ emissions from natural gas are slightly larger than in non-EJCs. Small amounts of biomass emissions appear only in EGUs located in non-EJCs.

$SO_2$ emissions trajectories between EJCs and non-EJCs are similar, with two exceptions (Fig 8). First, in EJCs, $SO_2$ emissions during 1997 and 1999 were higher in. Second, starting in 2003, $SO_2$ emissions decreased in EGUs in both types of communities but reductions were

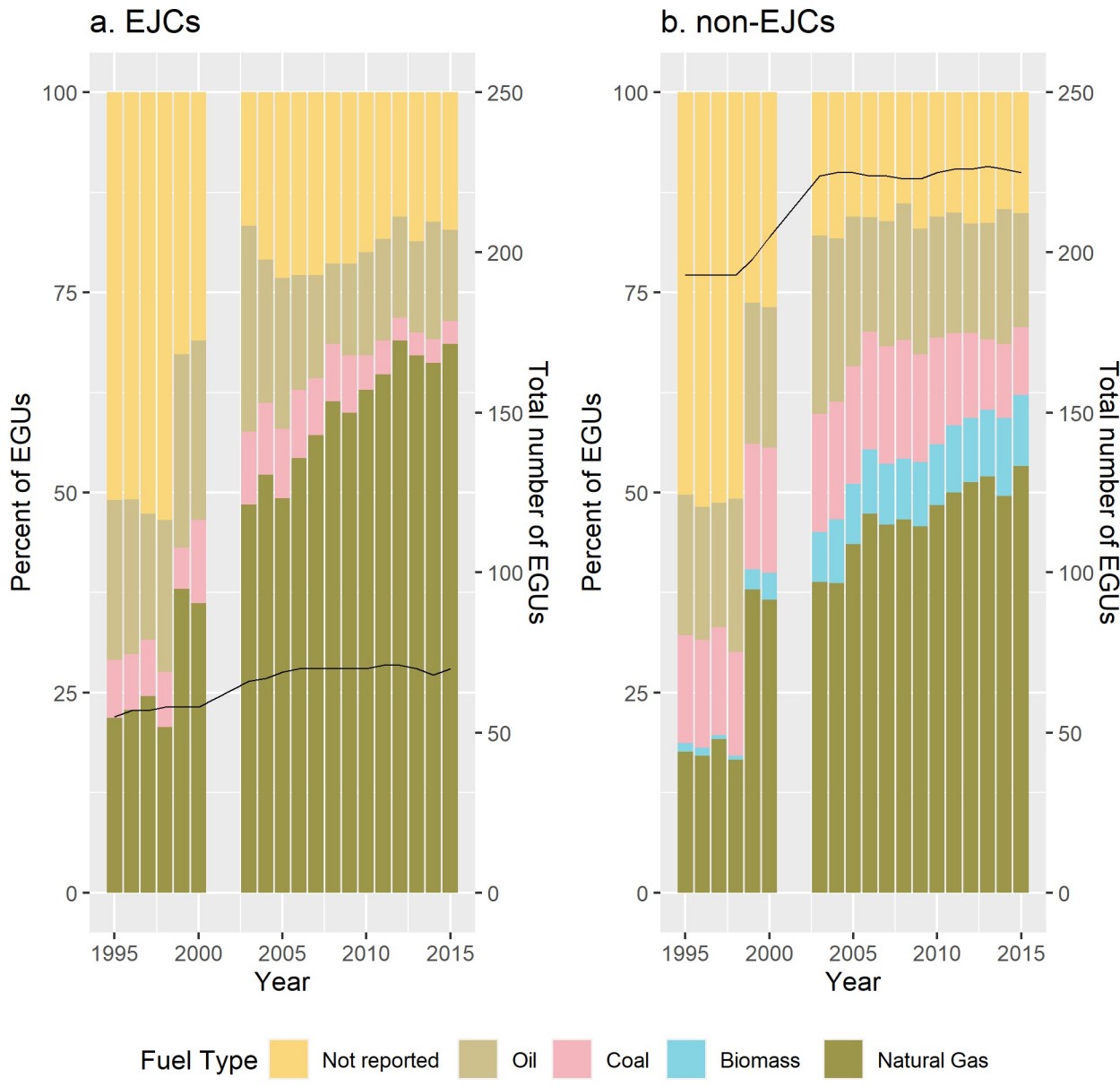

**Fig 4.** EGUs by fuel type in RGGI sited in (A) EJCs and (B) non-EJCs. Percent of total EGUs by fuel type is indicated in left y-axis. Black lines indicate the total number of EGUs and correspond to the right y-axis.

larger in EJCs especially after 2007. Since 1995, average $SO_2$ emissions have been mostly from oil and coal and decreased in both types of communities (but emissions dropped more in EJ communities after 2009).

NOx emissions between EJ and non-EJ communities are similar, but larger on average in EJ communities during 1995–2005 (Fig 9). After 2005, differences are due to biomass NOx emissions from EGUs in non-EJCs, as there are no biomass NOx emissions in EJCs. In both types of communities, the share of NOx emissions from coal, oil, natural gas dropped markedly after 2000, and with the exception of years 1999 and 2000, NOx emissions from biomass burning—only found in non-EJCs—remained a small fraction of average annual NOx emissions.

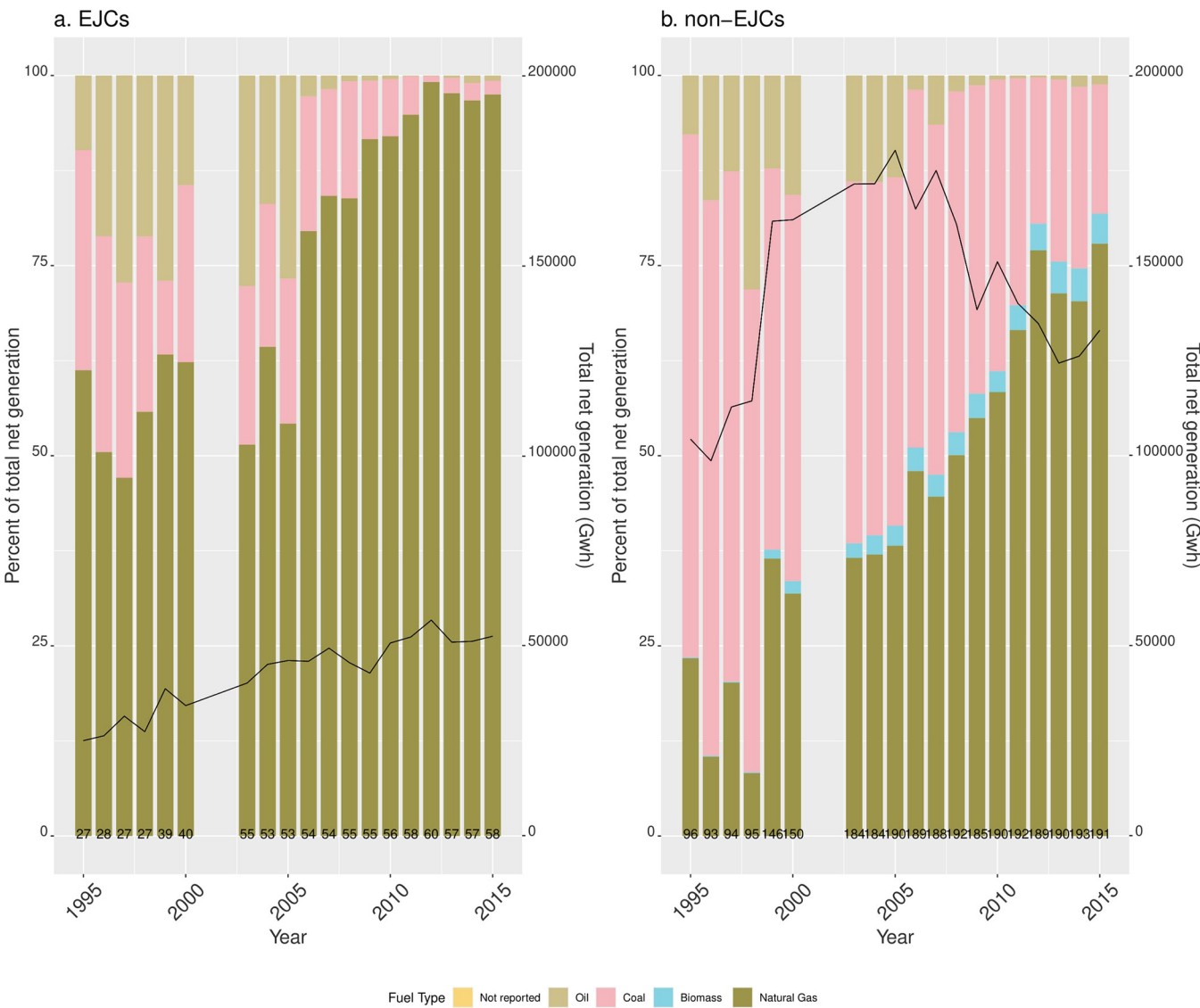

**Fig 5.** Total net generation by fuel type in EGUs in RGGI sited in (A) EJCs and (B) non-EJCs. Percent of total net generation by fuel type is indicated in left y-axis. Black lines indicate total generation and correspond to the right y-axis. EGUs that did not report fuel type also did not report generation and are thus omitted. The numbers at the bottom of the bars indicate the number of reporting units for each year.

## Discussion

Our analysis constitutes the first systematic and empirical attempt to assess disparities in pollutant potential from electricity production between EJ communities and non-EJ communities in RGGI states. There are a few key insights from our analysis. First, our proximity analysis shows that in states that participate in RGGI, a larger cumulative share of environmental justice populations live in proximity to electric power plants compared to non-EJ populations at similar distances. This is the case for distances of up to 39.4 miles from power plants in the comparison between Whites and people of color, and up to 20.5 miles for the poverty status comparison. However, the largest differences in population proximity occur between 0–6.2 miles based on race (up to 23.5% higher for people of color), and between 0–4.9 miles based

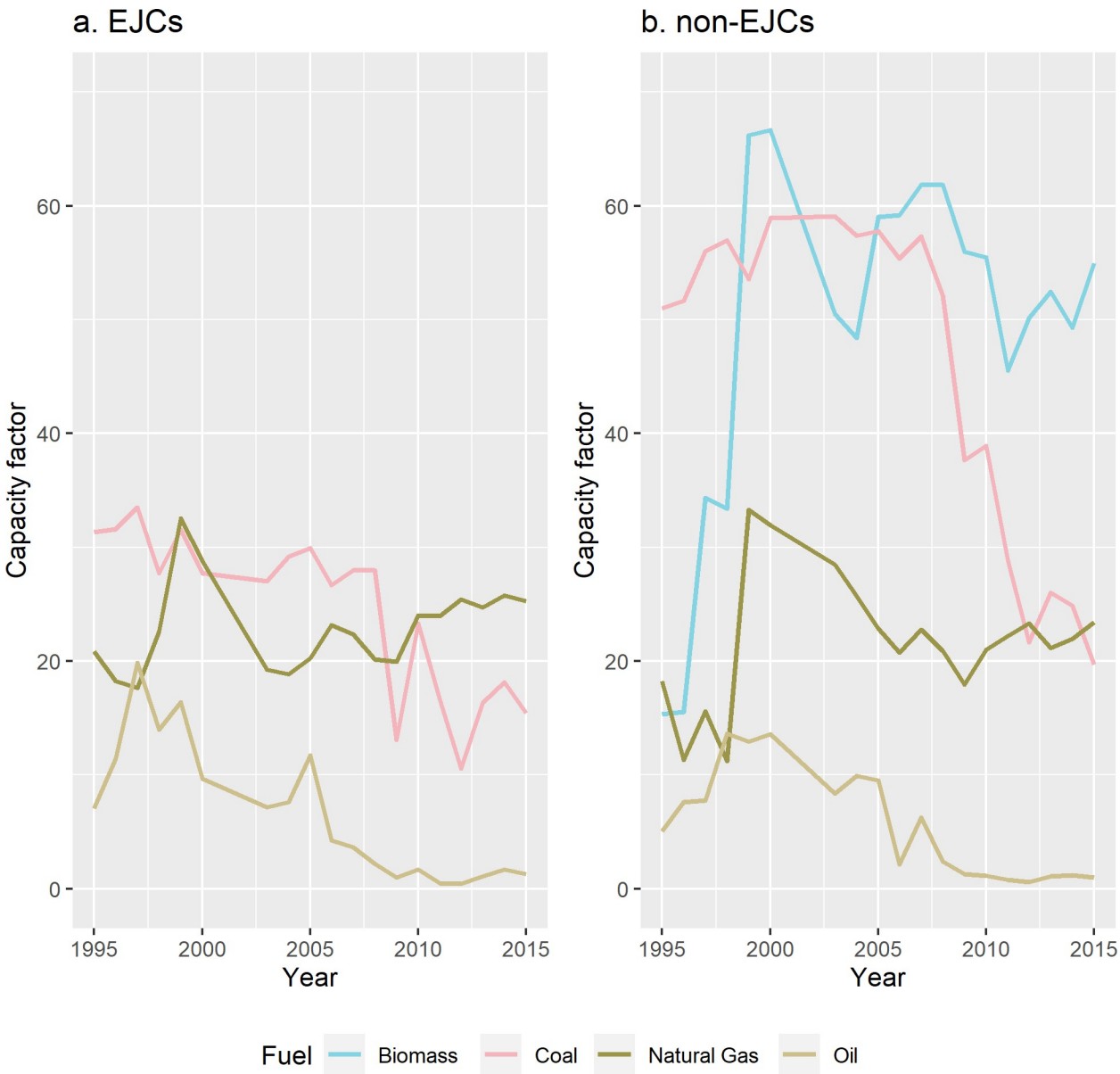

**Fig 6.** Mean capacity factor by fuel type in EGUs in RGGI sited in (A) EJCs and (B) non-EJCs. EGUs that did not report fuel type also did not report capacity factors and are thus omitted.

on income (up to 15.3% higher for people living in poverty). Furthermore, our siting analysis also found that 42.6% of EJ communities host between 2 and 5 EGUs, but only 28% of non-EJ communities host the same frequency range of EGUs. These results should be considered in energy sector emissions reduction policies that also contribute to reducing human health impacts from power plants because living within short distances of power plants is correlated to greater adverse health outcomes than for those people living farther away. For example, adverse birth outcomes have been found to be highest for populations living within 5 km or less from power plants emitting fine particulates [54]. While proximity to power plants is not sufficient to fully assess disparities in health impacts due to emissions from fossil-fuel

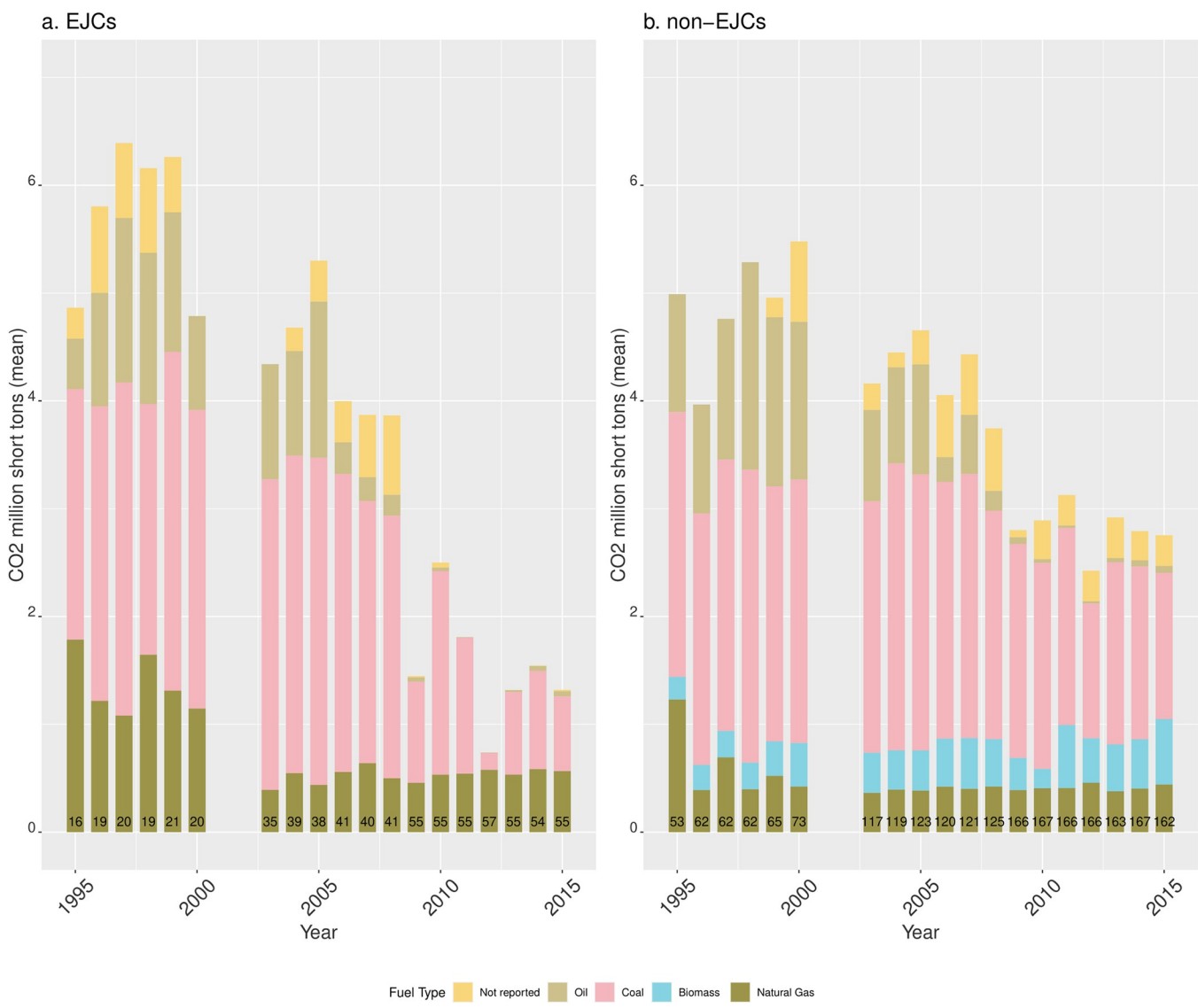

**Fig 7.** Mean annual $CO_2$ emissions by fuel type from EGUs in RGGI sited in (A) EJCs and (B) non-EJCs. The numbers at the bottom of the bars indicate the number of reporting units for each year.

electricity production, our results further validate the inequity that people living with low incomes and people of color continue to live near pollution emitting facilities and with higher potential exposure burdens due to a higher frequency of multiple facilities many decades after the problem of siting disparities first came to light [27].

Second, the coal-to-natural gas transition was more marked in environmental justice communities. As coal-fired EGUs became a smaller share of all EGUs between 1995–2015, natural gas in EJ communities became an increasingly larger fraction of all EGUs (and of total net generation) with a consistent upwards trend in both indicators of polluting potential. However, the time period also coincides with a general downwards trend of average emissions of CO2, $SO_2$, and $NO_x$ across EGUs in both types of communities (our third main finding). Finally, our fourth main finding is that coal-fired EGUs in non-environmental justice communities consistently run at higher rates than in environmental justice communities.

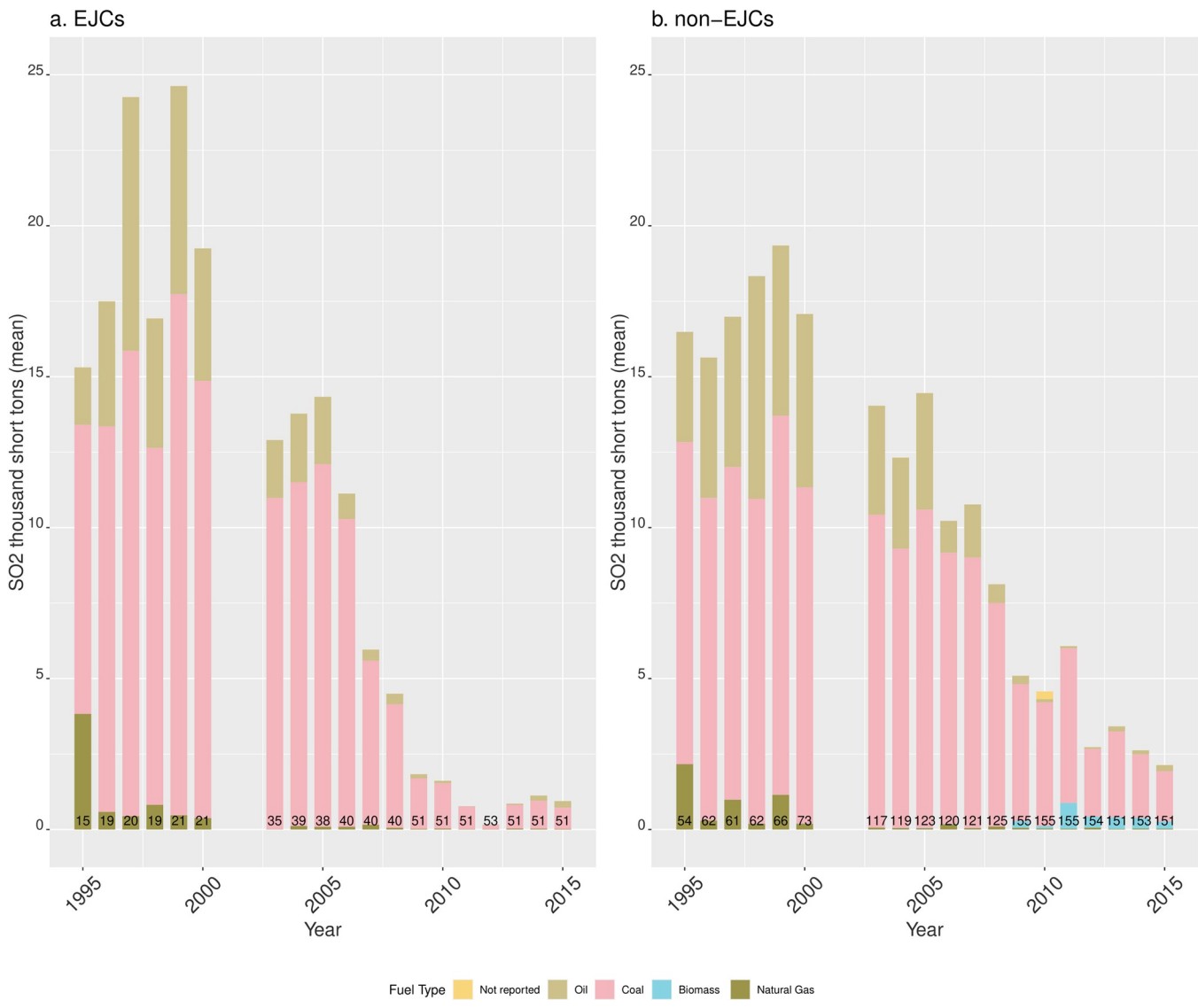

**Fig 8.** Mean $SO_2$ emissions by fuel type from EGUs in RGGI sited in (A) EJCs and (B) non-EJCs. The numbers at the bottom of the bars indicate the number of reporting units for each year.

## Comparison of our study to the EJ assessment of California's cap-and-trade program

Our study compares to and diverges from the only other empirical assessment, to our knowledge, of environmental justice burdens in a US regional GHG emissions market, focused on the first three years since California's cap-and-trade program began [55]. That study found that facilities regulated under the trading program are disproportionately located within environmental justice communities, that GHG emissions increased in more than half of regulated facilities since trading started, and that neighborhoods that saw GHG and co-pollutant emissions increases were largely low-income communities of color also with high rates of other socio-demographic markers associated with disadvantage. In comparison, two of our findings stand out. First, in RGGI states, environmental justice communities live closer than other populations to power plants. Second, the relative frequency of multiple EGUs in any one Census

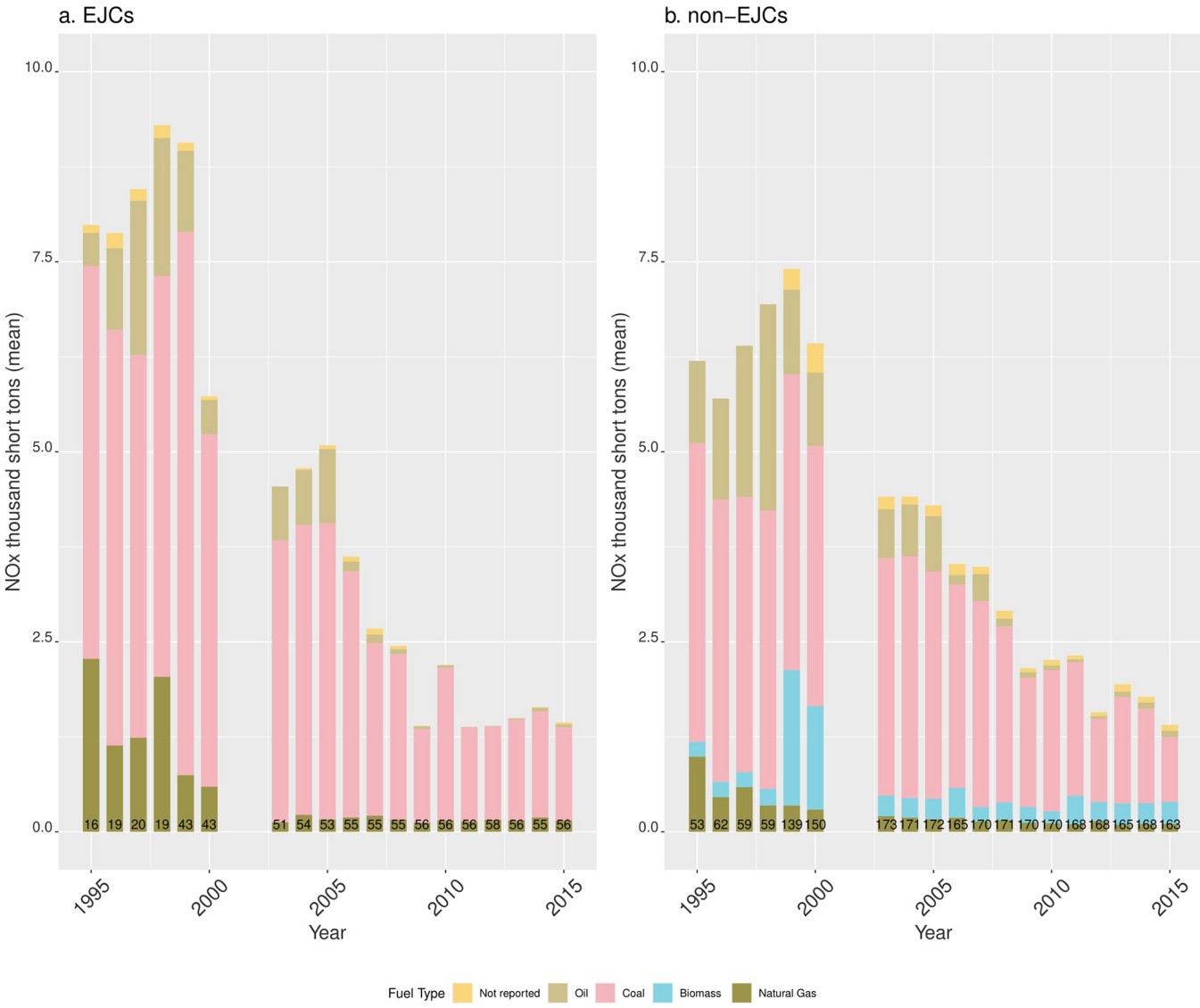

**Fig 9.** Mean NOx emissions by fuel type from EGUs in RGGI sited in (A) EJCs and (B) non-EJCs. The numbers at the bottom of the bars indicate the number of reporting units for each year.

Tract is higher in environmental justice communities than in non-environmental justice communities. These findings suggest that, similar to California's cap-and-trade program, environmental justice communities in RGGI are more exposed to dangerous co-pollutants compared to other communities. Our study diverges from the California assessment because, due to data limitations, we did not assess emissions from PM2.5 or air toxics, both of which have important implications for human health. In addition, our study was restricted to power sector emissions regulated under RGGI, whereas California's program covers all industrial sectors except for agriculture. But we were also able to look at a longer trajectory of emissions from the mid-1990s to 2015, which arguably were influenced by market and regulatory factors underway since before the start of the program in 2009. In our inductive assessment, we did not find changing trends in trajectories of $CO_2$, $NO_x$, or $SO_2$ specifically tied to the pre- and post-2009 years. Instead, emissions trajectories in our study follow a generally downwards trend

detectable since 1995 that is likely due to the transition from coal to natural gas, and perhaps to other mitigation policies in the power sector.

## Limitations of our research

As a first step in assessing disparities in power plant emissions exposure, our research has some limitations. First, our analysis does not consider dispersion of co-pollutants, as we focused on disparities in power plant characteristics in EGUs sited in EJ vs. non-EJ communities. Therefore, our analysis does not indicate exposure from the fate and transport to pollutants. The co-pollutant data used in our study ($SO_2$ and NOx) represent stack emissions, and do not consider how physical and photo-chemical atmospheric interactions create or disperse pollutants dangerous to human health or how power plant operating conditions differ among environmental justice and other communities. These interactions should be resolved using atmospheric chemistry and dispersion models that also include the dispersion of methane and fine particulates, which are outside the scope of the present research. Furthermore, we have considered power plant emissions in isolation from the cumulative environmental and health threats with which environmental justice communities are often overburdened, including burdens from transportation emissions. Our analysis is focused on emissions, siting, generation, and capacity factors of power plants hosted in communities in RGGI states. It has been argued that not comparing host vs. non-host communities results in underestimating environmental inequities [56]. We recognize that our results are conservative in this regard, but have shown that communities hosting at least one RGGI power plant do not have significantly higher rates of poverty or of people of color than non-host communities. In addition, EJ communities often endure burdens from energy production infrastructure such as shale gas expansion, oil trains, gas pipelines, and liquid natural gas export stations, but these are outside the scope of our work. For these reasons, our analysis underestimates the true burden of power plant-related exposure between environmental justice communities and other communities, which remains a limitation of current environmental justice assessments and policy [57].

While we have shown the transition in RGGI from coal to natural gas—underway since before the formal start of the program in 2009—our analysis did not allow us to establish if the steeper increase in natural gas units in environmental justice communities occurred due to conversion of coal units to gas, or if it entailed the construction of new plants. In addition, it was not possible for us to assess differences in the coal-to-gas transition between urban areas in RGGI states (that tend to be predominantly populated by people of color) and rural areas (that tend to be whiter).These are a key areas for future research that would also need to take into account permitting process dynamics among regulators and utility operators to determine the degree of disparity in new EGU construction in environmental justice communities.

Our results potentially underestimate the extent of locational disparities. In our analysis we use the federal poverty threshold to identify people living in poverty. However, the federal poverty methodology was developed in the 1960s and is recognized as outdated with respect to present family income needs. For example, research suggests that on average, families require twice the federal poverty level to meet basic needs [58]. Furthermore, the utility of the federal poverty level is limited because it does not take into account differences in cost of living across geographies in the United States. Furthermore, U.S. Census data reporting limitations on the ratio of income to poverty levels did not allow us to separate those living *above* the poverty line from those living *at* the poverty line. In our research, it would have been preferable to combine the poverty level of people living under the poverty level together with those living at the poverty level as our indicator of poverty, but this was not possible.

Beyond fuel type and generating capacity, there may be other factors that affect emissions levels which were not explored in this paper. Small emitters (i.e. with capacity rating <25MW) fall outside the scope of RGGI, and often have lower stack heights that can potentially contribute to localized dispersion of pollutants. Do these emitters constitute un-quantified emissions burdens in environmental justice communities? This question, to our knowledge, has not been explored systematically. There are outstanding issues of procedural justice that need to be considered in future research as well. Research has found deficits in environmental enforcement actions under the Clean Air Act and other environmental regulations in US counties with low-income populations when compared to other counties [59]. To what extent has there been equity in enforcement and compliance of power sector-related environmental regulations in environmental justice communities in RGGI states? These are critical questions to answer to deepen our understanding of disparities in environmental hazards and outcomes in environmental justice communities.

Our definition of what constitutes an EJ community identifies the intersection of populations of color with those of low income, but with the exception of the proximity analysis, our research does not consider power plant co-pollutant burdens in white EJ communities nor in wealthier communities of color. Other functional definitions of EJ communities may be more appropriate in other geographical contexts or reveal additional insights, as suggested by the five communities with high rates of people of color or people living in poverty that we classified as non-EJ community outliers (Fig 3). A guiding principle of environmental justice community organizing is to "let people speak for themselves" [60], which could imply that our definition of EJ communities may exclude communities that self-identify as EJ communities. While we find our method to be useful to systematically assess power plant-related burdens across a large number of communities, we recognize its potential to overlook communities burdened with historical and current environmental injustices, and invite further dialog among EJ advocates, researchers, and environmental regulators to refine the methods, and qualitative and quantitative data useful to assess the EJ status of communities.

## Conclusions

We have taken an inductive approach to answering questions from environmental justice advocates regarding power plant emissions burdens, focusing on the mid-1990s to 2015. During that period, multiple policies as well as increased availability of natural gas, among other reasons, have contributed to reshaping the landscape of fossil fuel burning for electricity production nationally in the US, but also in RGGI states. In our research, the clearest signal of change in emissions trajectories is that of $CO_2$, $SO_2$, and $NO_x$ from coal, which coincides with the "dwindling role for coal" [61] transition in the U.S. energy sector towards natural gas. But that transition also clearly entailed a large increase in both the total number of natural gas-firing units and in total net generation from natural gas in environmental justice communities, indicating that RGGI's focus on sector-wide, aggregate reductions has not explicitly considered potential impacts or inequitable burdens on environmental justice communities. This is especially salient because environmental justice advocates have warned that achieving equity and environmental justice goals requires explicit attention in carbon and climate mitigation policy, and should not be left up to chance in climate change mitigation policy [5]. In addition, concerns remain about the potential for carbon market forces to drive increases in carbon and co-pollutant emissions as utilities pursue lower-cost emissions reductions [3]. Some claim there is no evidence that emissions trading programs have exacerbated pollutant burdens in environmental justice communities [see 62–65]. But as Pastor et al. [29] have argued, these claims are largely theoretical and mostly lack empirical evidence [but see 66 for a notable

exception]. We note that others have suggested that factors other than RGGI market incentives have driven emissions reductions: lower natural gas prices have been found to be "the main driver of the RGGI system" [67], but teasing out the endogenous or exogenous reasons for aggregate emissions reductions in RGGI is outside of the scope of our research and remains an area for future research especially in regards to environmental justice burdens. The electric power sector in RGGI is a managed policy landscape, and moving forward, there need to be policies that bring benefits to overburdened communities. While we did not analyze RGGI's specific policies, we have demonstrated that differences in siting, generation, capacity factors, and emissions between environmental justice and non-environmental justice communities hosting RGGI power plants are significant (even if, as we have noted above, our analysis underestimates the true burden of power plant pollutant exposure).

The coal to natural gas transition has occurred in the context of a largely successful effort from environmental, clean and renewable energy, and climate change mitigation advocacy groups to retire coal-burning power plants to reduce GHG emissions and enable the transition to renewable energy. In our analysis, the disaggregate benefits of that transition—as measured by inequities in number of natural gas-fired, units, and generation of electricity and $CO_2$ emissions from natural gas—appear to have gone to non-environmental justice communities. But we have also shown that in RGGI states, non-EJ communities contain larger fractions of coal-fired EGUs than EJ communities, that reductions in average $CO_2$ emissions between 1995–2005 in EJ communities were slightly larger, $CO_2$ reductions in EJ communities were larger than in non-EJ communities after 2009, and average NOx emissions were larger in EJCs from 1995–2005, but dropped markedly after 2000. These results suggest that environmental inequalities manifest in many different ways. However, for environmental justice concerns, the most salient feature of the coal-to-natural gas transition shown in our analysis has been that the percent of natural gas-fired EGUs in EJ communities was consistently higher than in non-EJ communities.

While there have been aggregate reductions in $CO_2$, $SO_2$, and $NO_x$ from power plants, communities hosting power plants in RGGI states have divergent historical and current conditions of social and environmental disadvantage, and policies cannot operate as if race, ethnicity, and income do not matter because, as we have demonstrated, they clearly do. But which policy, market, or other forces can account for the faster ramp up of natural gas power plants in environmental justice communities? To what degree did the transition entail new natural gas facilities versus coal plant conversions? How did power plant pollution control technologies over time contribute to changes in emissions? To what extent have advocacy groups' campaigns aligned more with non-EJ communities, and how can these efforts help explain the differential patterns we found? These are critical questions to answer in light of inequities in enforcement of environmental, civil rights, and public health laws and policies, as well as pervasive discriminatory zoning and land-use practices, flaws in industrial risk assessment, and exclusionary practices that limit meaningful participation of environmental justice communities in decision-making, all of which prompted the emergence of the environmental justice in the first place [68, 69].

Our analysis demonstrates that, in the absence of directed policy attention, there are real differences in siting and operation of facilities in communities of color and low-income communities, and the polluting potential for overburdened communities is higher. Therefore, policy efforts to address the energy generation sector's emissions need to directly consider the equity implications of policy options, not just their impact on one or more pollutants. And, despite the fact that $CO_2$ may be globally impactful, local effects of energy production cannot be ignored, ethically or legally.

## Supporting information

**S1 Fig. Percent people of color and percent in poverty statistics in k-means cluster analysis of all census tracts in RGGI states.**
(TIF)

**S1 Table. Diagnostic statistics of difference of means of polluting potential indicators between EGUs in environmental justice communities and non-environmental justice communities in RGGI states.**
(XLSX)

## Acknowledgments

We would like to thank Mary Pham for providing research support in the early stages of research development. We consulted throughout this research with environmental justice advocates and experts working in grassroots organizations. Their guidance and expertise greatly helped us to better understand community perspectives. These experts chose not to be individually acknowledged for their part in this research.

## Author Contributions

**Conceptualization:** Juan Declet-Barreto, Andrew A. Rosenberg.

**Data curation:** Juan Declet-Barreto.

**Formal analysis:** Juan Declet-Barreto.

**Funding acquisition:** Andrew A. Rosenberg.

**Investigation:** Juan Declet-Barreto, Andrew A. Rosenberg.

**Methodology:** Juan Declet-Barreto.

**Project administration:** Juan Declet-Barreto.

**Software:** Juan Declet-Barreto.

**Supervision:** Andrew A. Rosenberg.

**Validation:** Andrew A. Rosenberg.

**Visualization:** Juan Declet-Barreto.

**Writing – original draft:** Juan Declet-Barreto, Andrew A. Rosenberg.

**Writing – review & editing:** Juan Declet-Barreto, Andrew A. Rosenberg.

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
