## [Decision Letter · Decision Letter 0]

31 Jan 2022

PONE-D-21-29053Exploratory analysis of changing power plant pollutant burdens on environmental justice communities in the Regional Greenhouse Gas Initiative statesPLOS ONE

Dear Dr. Declet-Barreto,

Thank you for submitting your manuscript to PLOS ONE. After careful consideration, we feel that it has merit but does not fully meet PLOS ONE’s publication criteria as it currently stands. Therefore, we invite you to submit a revised version of the manuscript that addresses the points raised during the review process.

 Please see detailed reviewer reports below. The reviewers raise some concerns about bias and clarity in the methodology. If you need more time to address these than typically provided for a revision decision, you may reach out to the journal office for additional time.

We look forward to receiving your revised manuscript.

Kind regards,

Hanna Landenmark

Senior Editor

PLOS ONE

Journal Requirements:

“This work was funded by the Kendall Fellowship from the Union of Concerned Scientists for the first author.”

“Juan Declet-Barreto's contribution came from work completed while a Kendall Science Fellow at the Union of Concerned Scientists (UCS), which is funded by unrestricted dollars allocated by the UCS Board.”

“Juan Declet-Barreto's contribution came from work completed while a Kendall Science Fellow at the Union of Concerned Scientists (UCS), which is funded by unrestricted dollars allocated by the UCS Board.”

Reviewers' comments:

Reviewer's Responses to Questions

**Comments to the Author**

1. Is the manuscript technically sound, and do the data support the conclusions?

Reviewer #1: Partly

Reviewer #2: Partly

Reviewer #3: No

Reviewer #4: Partly

2. Has the statistical analysis been performed appropriately and rigorously? 

Reviewer #1: I Don't Know

Reviewer #2: Yes

Reviewer #3: No

Reviewer #4: No

3. Have the authors made all data underlying the findings in their manuscript fully available?

Reviewer #1: No

Reviewer #2: Yes

Reviewer #3: No

Reviewer #4: No

4. Is the manuscript presented in an intelligible fashion and written in standard English?

Reviewer #1: Yes

Reviewer #2: Yes

Reviewer #3: Yes

Reviewer #4: Yes

5. Review Comments to the Author

Reviewer #1: - More suitable title should be selected for the article. Title should decrease to 10-12 words.

- The abstract should state briefly the purpose of the research, the principal results and major conclusions. An abstract is often presented separately from the article, so it must be able to stand alone.

- It is suggested to present the structure of the article at the end of the introduction.

- The major defect of this study is the debate or Argument is not clear stated in the introduction session. Hence, the contribution is weak in this manuscript. I would suggest the author to enhance your theoretical discussion and arrives your debate or argument.

- It is suggested to compare the results of the present research with some similar studies which is done before.

- More suitable title should be selected for the figure 4 instead of “Demographics in k-means cluster analysis of Census Tracts hosting EGUs in RGGI.”.

- It is suggested to add articles entitled “Okeke et al. City as Habitat; Assembling the Fragile City”, “Gibergans-Baguena et al. The Quality of Urban Air in Barcelona: A New Approach Applying Compositional Data Analysis Methods” and “Angelevska et al. Urban Air Quality Guidance Based on Measures Categorization in Road Transport” to the literature review.

- Page 11: the following paragraph is unclear, so please reorganize that:

“In addition, others have voiced more fundamental problems with carbon trading that do not center on emissions reductions, namely that carbon markets enable commodification of air pollution (30), effectively giving polluting facilities private property rights over the atmosphere.”

- DOI of the references must be added (you can use “" ext-link-type="uri" xlink:type="simple">https://crossref.org/").

- Much more explanations and interpretations must be added for the Results, which are not enough.

- Please make sure your conclusions' section underscore the scientific value added of your paper, and/or the applicability of your findings/results, as indicated previously. Please revise your conclusion part into more details. Basically, you should enhance your contributions, limitations, underscore the scientific value added of your paper, and/or the applicability of your findings/results and future study in this session.

Reviewer #2: Line 39: you need to define EJ for the ease of understanding before using it.

Lines 88-93: There is a repetition of a sentence. Please remove it.

Line 138: You have defined EJCs so you can use it here rather than using the complete name. Otherwise, there is no point of defining it earlier.

For population proximity, I would recommend generating a map of proximity using all the power plants to show the spatial heterogeneities in census tracts rather summarizing them into a line graph. Similarly, the maximum number for distance in Figure is 10 miles. Does this implies that no population/community is present beyond 10 miles proximity to power plants? I think a map is required to explain such ambiguity.

For the identification of EJ communities, more details are required to clearly state what EJ and Non-EJ community in this study means. Is an EJ community with larger ethnic composition and poverty? or is it the opposite. Please clearly define these communities.

For polluting potential of EGUs, it is strongly recommended to also check the significance of increasing or decreasing trends between 1995-2015 in the context of RGGI.

Lines 305-309: If the data for 2001 and 2002 is not used, how can one determined if the communities saw he "Sharpest" increase?

Line 316: Generation of what? Please clarify for the sake of readers

Line 370: there is a typo

In discussion, authors state that one of the key insights from this study is that EJ populations live closer to electric power plants. From where the authors deduce this insight? is it from Figure 3? if yes, the figure fails to explain which curves among the four are EJs and which ones are non-EJs.

Similarly, Authors describe that validation of inequity that low income and people of color continue residing closer to pollution emitting facilities. In Figure 3, 40% cumulative population for all four curves is at a distance of ~4 mile and the rest of the population is beyond this. How the threshold of "closer" is defined in this study? and How to decide if 4 miles proximity is having health impacts? Similarly, did the authors use the temporal data on population (1995-2015) as well? if yes, you need to mention it. If not, how is it deduced that populations are still living closer to these facilities even after decades (line 389). There might be a decrease in the population residing the proximity of these facilities if one look at the temporal trends of population. However, it might require precise gridded population data to evaluate it. In a nutshell, the proximity section in this manuscript needs further explanations to clarify the issues.

For the comparison of EJ- and Non-EJ-communities, I would recommend using some systematic analysis (i.e., ANOVA) to evaluate if the transitions were significantly different as stated in Lines 392-393.

lastly, while the study offers a good discussion and details on limitations, there is still a lack of focus on what are the implications of the results and how the environmental planners along with community stakeholders might benefit from the findings of this study. Hence, it is recommended to discuss the potential implications of the findings from the perspective of decisions and policy.

Reviewer #3: This study addresses a vital question: whether market-based greenhouse gas emission reduction mechanisms such as RGGI increase environmental injustice. But it suffers from a fatal flaw in methodology and also from a lack of effort to explain the patterning of findings. The authors selected on the dependent variable (by only including the 285 Census tracts hosting one EGU, rather than all Census tracts in the 11 RGGI states). This has biased the analysis, which only compares host communities with host communities; non-host communities (that tend to be whiter and higher in income) are entirely left out of the analysis which results in under-measuring environmental inequality (see Mohai 1995, his famous critique of Anderton, Anderson, Oakes and Fraser 1994).

Mohai, Paul. "The demographics of dumping revisited: examining the impact of alternate methodologies in environmental justice research." Virginia Environmental Law Journal (1995): 615-653.

The authors demonstrate a good working knowledge of environmental justice issues; if they choose to redo the analysis to eliminate bias, more could be done to increase this study’s contribution to the scientific literature. Specifically, the paper suffers due to lack of explanation of the findings: what is driving the shift? The authors would need to at least try to explain the patterning of environmental inequalities found so that the paper can be stronger scientifically and also be a better contribution for crafting policy.

Due to the methodological problems, I am recommending that the paper be rejected. But I include detailed comments below in case the authors decide to re-do the analysis:

Comments on Introduction and Literature Review:

1. There needs to be some explanation of why RGGI matters for environmental inequality: how is RGGI hypothesized to decrease or increase environmental burdens for EJCs? How, if at all, might RGGI change the market calculation of what type of power plant might be profitable? The authors cite a few papers on these questions but I know there are many more.

Comments on data and variables:

1. It was not clear why “prime movers” mattered for this analysis. Do they have an effect on emissions? If so, how? If prime mover is just another way of referring to fuel, the authors should omit that phrase.

2. On Page 10-11, the authors don’t provide a clear definition of “environmental justice community.” Does this mean census tracts above a certain threshold of percent people of color AND above a certain threshold of people in poverty (or was it OR a certain threshold of people in poverty)? And what were “non-EJC outliers?”

3. On page 21, describing the limitations of the study, the authors state that it would have been preferable to consider people living under the poverty level (presumably, for a comparison with those living at poverty level), but due to limitations in Census data this was not possible. Given that the poverty level is such an outmoded undercount of people living in poverty, I doubt that comparing people in poverty with people in deep poverty would have been as useful as comparing people in poverty with those better off. But the authors could have compared people at 100% of poverty level with people at 200% of poverty level (the near-poor). These data are available in Table POV01, which should be available for the years up to 2014.

Comments on methods:

1. Unfortunately, the methods used in the analysis were fatally flawed: the authors selected on the dependent variable by only including the 285 census tracts that host EGUs in the analysis. This has biased the analysis because it compares host communities with host communities, leaving non-host communities entirely out of the analysis. The scope of the analysis is very important, as this methodological choice has been proven to underestimate environmental inequality (see Mohai 1995).

Aside from this serious problem, other methodological choices were not adequately explained:

2. It is not clear why the distance of four miles was chosen to represent proximity to EGUs. Is there some scientific justification for defining proximity in terms of four miles from the centroid of each census tract, rather than one mile, two miles, etc.?

3. The authors seemed to miss out on some of the precision with which distance-based measures can find environmental inequalities. Did people of color or people in poverty live closer to EGUs than others? Or was a four-mile radius the basis for the analysis?

Comments on interpretation of findings:

1. The authors should at least try to find explanations for these patterns, such as the finding that there were fewer coal-fired EGUs and more natural gas-fueled EGUs in EJ communities. Below are some possible explanations that could be explored:

• Natural-gas power plants may be newer than coal-fired power plants, and thus may have been sited recently. Since 1990 there has been more opposition to siting environmentally hazardous facilities; non-EJ communities may have successfully blocked new power plants, leaving EJ communities “the path of least resistance.”

• This also could be driven by zoning: it is well established that minorities and the poor are more numerous in places zoned for industrial use.

• Investment in new (natural gas) power plants or in the conversion of outmoded ones may have occurred in more densely populated urban and metropolitan areas versus rural areas. Populations in rural areas tend to be whiter and poorer.

• It is likely that there is a lot of variation in the patterns of proximity to EGUs in RGGI-participating states: states that are white, rural and poor with little investment in new infrastructure (e.g., Maine) may be the same states with more outmoded coal-fired plants; states that are urban and diverse with extremes of wealth and poverty (e.g., New Jersey) may have more natural gas plants.

• If any of these explanations are correct, they might explain the finding that CO2 and SO2 emissions in EJCs were higher between 1995-2005 but reductions were larger in EJCs after 2007.

2. The finding regarding emissions appears to be a bright spot: emissions have gotten lower since 2005 for EJCs. But the authors needs to do more to illuminate an understanding of this finding: did the pattern of emissions change due to the shift from gas to coal? Was it due to plant modernization (better pollution controls, cleaner fuels, or both)? Was it because methane emissions increased while CO2 and SO2 decreased? All of these questions are really crucial ones for redressing environmental inequalities.

3. Finally, what is the significance of the timing of the shift? Could it be due to policy (Energy Policy Act, Clean Power Plan, etc.)? Or the economy (gas prices)? If finding answers is beyond the scope of this study, the authors could speculate on these questions and suggest future research.

Comments on Conclusions:

1. The authors should mention the impacts on EJCs from the expansion of shale gas and oil, including fracking wells, oil trains, gas pipelines, liquid natural gas export terminals and compressor stations, among others.

Comments on Figures:

Figure 2: I was unable to understand from the manuscript why “prime mover type” was important. If it was because it affects the amount or toxicity of emissions, this should be made clear. I am not sure what this figure adds; if space limitations become an issue, I recommend omitting it.

Figure 3: what are the “outliers”?

Figures 5 – 10: There are too many figures, which take up space the authors need.

Mohai, Paul. "The demographics of dumping revisited: examining the impact of alternate methodologies in environmental justice research." Virginia Environmental Law Journal (1995): 615-653.

Reviewer #4: Title: Exploratory analysis of changing power plant pollutant burdens on environmental justice communities in the Regional Greenhouse Gas Initiative states.

Summary of paper: While there have been many studies that evaluate the spatial distribution of environmental hazards to environmental justice populations (predominantly characterized as low income communities and communities of color), few studies have evaluated environmental justice burdens in GHG emissions markets. This study contributes to environmental justice literature by assessing pollution and power plant disparities in RGGI states. Authors found that a larger share of environmental justice populations live near power plants than non-ej populations and that the coal-to-natural gas transition was more evident in environmental justice communities. Reductions in CO2, NOx, and SO2 emissions were found to be similar in EJ communities and non ej communities over the time period studied, with a few differences.

This paper offers valuable contributions to the field and contextualizes the importance of these considerations for GHG emissions and broader climate policy while discussing limitations clearly. However, there are a few areas with needed clarification of methods, presentation and discussion of certain results, citations, and supporting figures.

Major changes:

Literature Review (e.g., Line 86): add/incorporate citation - McCoy E. Which came first, coal-fired power plants or communities of color? Assessing the disparate siting hypothesis of environmental justice (Doctoral dissertation).

Literature Review: add / incorporate citation - Diana, Ash, and Boyce (2021) Green for All: Integrating Air Quality and Environmental Justice into the Clean Energy Transition -- https://peri.umass.edu/economists/michael-ash/item/1408-green-for-all-integrating-air-quality-and-environmental-justice-into-the-clean-energy-transition

Lines 189-192: In this paragraph, there should be more clarification on the exact variables that were used from the Census to form the “percent people of color” and white variables in the analysis. Specifically, was non-Hispanic white used to identify white populations in the analysis? Was the “percent people of color” variable built so that certain minority populations were not double counted? i.e. if the race variables of the Census (African American alone, Asian alone, etc) were combined with the ethnicity variable, ALL who identify as Hispanic or Latino of any race, then people who identify as both African American AND Hispanic or Latino would be double counted in the population of color. The same would be true for Hispanic White individuals in both the white population and the people of color population. Further clarification is needed to confirm this error did not take place. If populations were double counted, this could significantly impact the analysis and require major revisions.

Lines 199 - 205: A figure depicting the methods of CDF would clarify the process followed for the reader

Line 242: The analysis drops to focus only on the 285 Census tracts that contain an EGU. It would be helpful to clarify how many total ej and non-ej tracts there were regardless of presence of an EGU in the study area and it seems as though some results are presented in a way that makes the distinction between presence of EGU or not unclear. It would aid the analysis to consider the rate of EGU in non-ej communities relative to ALL non-ej communities and the rate of EGU in ej communities relative to ALL ej communities.

Lines 232 - 246: it’s not clear what the threshold for defining an EJ community is and if there is no threshold, what makes an ej community distinct from a non-ej community.

Lines 291 - 298: the presentation of the results on the number of EGUs cited in ej and non ej tracts is not convincing. One possible clarification would be including the rate of EGUs in EJ or non EJ tracts relative to ALL ej or non EJ tracts in the study area, not just those with EGUs. Continued below:

Lines 412 - 413: given the results, the statement “environmental justice communities more frequently host multiple EGUs compared to non-environmental justice communities” seems inaccurate because non-ej communities still host a higher number of EGU units (49) than ej communities (20). Reading the data presented, a reader could interpret that a larger proportion of EGUs are in non-ej communities, both when sited as 1 unit or 2-5 units together -- this seems to be different from how this data is discussed by the authors. This result (above) and discussion of the result here should be removed, reworded/clarified, or considered from another perspective (i.e., as mentioned above, percentage of ej community tracts with EGUs / ej community tracts total vs percentage of non-ej community tracts with EGUs / non-ej community tracts total) in order to be supported by the data.

Minor changes:

Line 45: missing citation

Lines 88-90: cut, repeated verbatim in lines 91-93

Line 102: missing citation

Line 170: clarify the analysis is limited to plants greater than 25 MW. In the discussion, it is stated that above 25 MW are those that fall within the scope of RGGI, so it would be helpful to the reader if that was stated earlier in this spot as well.

Line 232 - remove word unbias. Any method requires choices that include bias.

6. PLOS authors have the option to publish the peer review history of their article (what does this mean?). If published, this will include your full peer review and any attached files.

Reviewer #1: No

Reviewer #2: No

Reviewer #3: **Yes: **Diane M. Sicotte

Reviewer #4: No

---

## [Author Response · Author response to Decision Letter 0]

4 Apr 2022

Please see response to reviewers.

---

## [Editor Report · Decision Letter 1]

24 May 2022

PONE-D-21-29053R1Environmental justice and power plant emissions in the Regional Greenhouse Gas Initiative statesPLOS ONE

Dear Dr. Declet-Barreto,

Thank you for submitting your manuscript to PLOS ONE. After careful consideration, we feel that it has merit but does not fully meet PLOS ONE’s publication criteria as it currently stands. Therefore, we invite you to submit a revised version of the manuscript that addresses the points raised during the review process.

I have served as a reviewer for the first draft of this manuscript. The article is mostly ready for publication, but minor revisions are needed because the authors’ interpretation of their findings in the Findings, Discussion and Conclusions sections are not consistent with the data presented in the article (see Comments below). Revisions on interpretation of findings are necessary for publication as is clarifying the discussion of findings on Page 18. The recommendation that the authors emphasize the disparity in the number of EGUs is suggested but not required.

Comments on data and variables:

In the revised version of the paper, the authors are no longer testing for disparities in the presence/absence of EGUs within a 4-mile radius, but rather testing for disparities in distance/proximity to EGUs. They are still comparing conditions only in host communities, but seeking to determine whether, within host communities, EJCs live closer to EGUs than non-EJCs. The methodology used (CDF function) was appropriate to the research question, and the researchers were able to provide convincing evidence of disparities in proximity.

Comments on guiding questions and their relationship to findings and interpretation of findings (Page 7):

Question 2 asks if income and race are an appropriate basis for delineating EJ communities from non-EJ communities.

Question 3 asks what is the distribution of the polluting potential as measured by fuel type, total net generation, capacity factor and actual pollution from CO2, SO2 and NOx in EJCs vs. non-EJCs.

Findings:

1. People of color are 23.5% more likely to live within 0-6.2 miles of an EGU; people living in poverty are 15.3% more likely to live within 0-4.9 miles of an EGU.

2. 42.6% of EJCs host 2-5 EGUs but only 28% of non-EJCs do.

This is a rather significant point, one that should be emphasized more when discussing possible disparities in exposures to air pollutants from EGUs.

3. Overall, from 1995-2015 % of coal-fired EGUs smaller in EJCs (higher in non-EJCs).

4. Since 2003: % of natural gas EGUs higher in EJCs than non-EJCs.

5. Total net generation in non-EJCs about 3 times as high as in EJCs (stated on Page 17).

Findings 3 and 5 provide another indication that RGGI has not affected all communities evenly, since non-EJCs face higher risk of exposure to pollutants from coal-fired power plants than EJCs, and pollution potential in terms of generation is higher for non-EJCs. It is higher in other ways for EJCs (EJCs closer to EGUs, more EJCs host multiple EGUs, EJCs suffer more potential exposure from gas-fired plants). The authors should mention and discuss all the ways in which environmental inequalities manifest.This implies that non-EJCs have benefited less from the shift from coal to natural gas than EJCs. This, too, should be mentioned and discussed.These results also imply that the definition of EJCs in purely demographic terms has some important limitations: it has the effect of making distributional inequalities that affect communities not defined as EJCs invisible, perhaps because their demographics are just slightly too white or not poor enough. The authors should at least mention this, perhaps couching it in terms of answering Question 2.

6. Between 1995-2005, CO2 emissions in EJCs were slightly higher; reductions were larger in EJCs after 2009.

This implies that EJCs might have benefited somewhat, or in some ways, from reductions in CO2 emissions due to RGGI. While there may be other factors that partially or totally eclipse this benefit, the authors should not ignore this finding.

7. After 2003, CO2 emissions from natural gas rose in EJCs more than in non-EJCs.

8. (Page 18): S02 emissions higher in both types of communities (or just EJCs)?

The authors should clarify the discussion on Page 18; as written, it is unclear.

9. Starting in 2003, S02 emissions decreased in EGUs in both types of communities but reductions were larger in EJCs especially after 2007.

As is true of #6, the authors need to discuss this finding and put it into context.

10. S02 emissions mostly from coal and oil; decreased more in EJCs after 2009.

Same as #6 and #9.

11. Average NOX emissions were larger in EJCs during 1995-2005 but dropped markedly after 2000.

Same as #6, #9 and #10.==============================

We look forward to receiving your revised manuscript.

Kind regards,

Diane Sicotte

Guest Editor

PLOS ONE
---

## [Author Response · Author response to Decision Letter 1]

15 Jun 2022

See the cover letter for our responses.

---

## [Editor Report · Decision Letter 2]

23 Jun 2022

Environmental justice and power plant emissions in the Regional Greenhouse Gas Initiative states

PONE-D-21-29053R2

Dear Dr. Declet-Barreto,

We’re pleased to inform you that your manuscript has been judged scientifically suitable for publication and will be formally accepted for publication once it meets all outstanding technical requirements.

Kind regards,

Diane Sicotte

Guest Editor

PLOS ONE
---

## [Editor Report · Acceptance letter]

27 Jun 2022

PONE-D-21-29053R2 

Environmental justice and power plant emissions in the Regional Greenhouse Gas Initiative states 

Dear Dr. Declet-Barreto:

I'm pleased to inform you that your manuscript has been deemed suitable for publication in PLOS ONE. Congratulations! Your manuscript is now with our production department. 

Kind regards, 

on behalf of

Dr. Diane Sicotte 

Guest Editor

PLOS ONE